# Impacts of COVID-19 on clinical research in the UK: A multi-method qualitative case study

David Wyatt[1,2]*, Rachel Faulkner-Gurstein[1,2], Hannah Cowan[1,2], Charles D. A. Wolfe[1,2]

1 School of Population Health and Environmental Sciences, King's College London, United Kingdom,
2 National Institute for Health Research Biomedical Research Centre at Guy's and St. Thomas' NHS Foundation Trust and King's College London, United Kingdom

* david.wyatt@kcl.ac.uk

**Data Availability Statement:** Data from this study take the form of interview transcripts, Hospital Trust and national documents, and observations of closed meetings. These data cannot be shared publicly, but extracts from interviews are presented

## Abstract

### Background

Clinical research has been central to the global response to COVID-19, and the United Kingdom (UK), with its research system embedded within the National Health Service (NHS), has been singled out globally for the scale and speed of its COVID-19 research response. This paper explores the impacts of COVID-19 on clinical research in an NHS Trust and how the embedded research system was adapted and repurposed to support the COVID-19 response.

### Methods and findings

Using a multi-method qualitative case study of a research-intensive NHS Trust in London UK, we collected data through a questionnaire (n = 170) and semi-structured interviews (n = 24) with research staff working in four areas: research governance; research leadership; research delivery; and patient and public involvement. We also observed key NHS Trust research prioritisation meetings (40 hours) and PPI activity (4.5 hours) and analysed documents produced by the Trust and national organisation relating to COVID-19 research. Data were analysed for a descriptive account of the Trust's COVID-19 research response and research staff's experiences. Data were then analysed thematically. Our analysis identifies three core themes: centralisation; pace of work; and new (temporary) work practices. By centralising research prioritisation at both national and Trust levels, halting non-COVID-19 research and redeploying research staff, an increased pace in the setup and delivery of COVID-19-related research was possible. National and Trust-level responses also led to widescale changes in working practices by adapting protocols and developing local processes to maintain and deliver research. These were effective practical solutions borne out of necessity and point to how the research system was able to adapt to the requirements of the pandemic.

### Conclusion

The Trust and national COVID-19 response entailed a rapid large-scale reorganisation of research staff, research infrastructures and research priorities. The Trust's local processes

within the body of the paper that make up the "minimal dataset."

**Funding:** DW, RFG, HC and CADW are all funded by the National Institute for Health Research (http://nihr.ac.uk/) Biomedical Research Centre at Guy's and St Thomas' NHS Foundation Trust and King's College London (Grant number IS-BRC-1215-20006). The funder had no role in study design, data collection and analysis, decision to publish, or preparation of the manuscript. The views expressed are those of the authors and not necessarily those of the NHS, the NIHR or the Department of Health and Social Care.

**Competing interests:** No

that enabled them to enact national policy prioritising COVID-19 research worked well, especially in managing finite resources, and also demonstrate the importance and adaptability of the research workforce. Such findings are useful as we consider how to adapt our healthcare delivery and research practices both at the national and global level for the future. However, as the pandemic continues, research leaders and policymakers must also take into account the short and long term impact of COVID-19 prioritisation on non-COVID-19 health research and the toll of the emergency response on research staff.

## Introduction

Clinical research is a core part of the global response to COVID-19. The United Kingdom (UK), with its research system embedded within the National Health Service (NHS), has been singled out by commentators globally for the scale and speed of its COVID-19 research response, particularly in terms of trial recruitment [1–3]. Reporting from within the UK context, Darzi et al. suggest that participating in clinical trials should be part of the clinical pathway for all COVID-19 patients [4]. To date, 95 nationally prioritised COVID-19 research projects, labelled Urgent Public Health studies, have commenced [5]. These and a large number of other COVID-19 studies have rapidly been set up and rolled out across UK hospitals. Supporting and facilitating such research has been made possible by the widespread reorganisation of the NHS' existing embedded research infrastructure. This reorganisation was initiated by the UK's Department Health and Social Care (DHSC), which on 16th March 2020 stated that all National Institute for Health Research (NIHR) funded staff should "prioritise nationally-sponsored COVID-19 research activity" [6]. They later clarified, stating "the NIHR Clinical Research Network is pausing the site set up of any new or ongoing studies at NHS and social care sites that are not nationally prioritised COVID-19 studies [6]." Such decisions were said to "enable our research workforce to focus on delivering the nationally prioritised COVID-19 studies or enable redeployment to frontline care where necessary [6]." To date, reports have focused on the outputs of this research, such as the outcomes of vaccine studies or results of treatment trials, and on frontline clinical staffing, healthcare provision and resource strains faced by hospitals and health care systems at national and global levels [7–12]. As yet, there has been no analysis of the organisation of the research response and the broader impact of the reorganisation of hospitals and research facilities that has allowed clinical research and emergency care work to take place during the pandemic.

In this paper we provide a detailed exploration of how the embedded research infrastructure in one NHS Trust in South London. Throughout this paper, we e use the pseudonym South London Acute Trust (SLAT) to avoid direct identification. This Trust was repurposed to support the completion of COVID-19 research and delivery of frontline care. SLAT is one of the UK's most research-active Trusts, annually recruiting over 19,000 patients to more than 550 studies. Between February and December 2020, SLAT opened over 80 COVID-19 studies, with more than 18 of these classed as Urgent Public Health studies, recruiting over 7,000 participants. Within this context, we ask: what have been the impacts of COVID-19 on SLAT's clinical research system, and how has the embedded research system been adapted and repurposed to support the COVID-19 response?

Prior to the pandemic, the process of setting up and managing a clinical research study within a UK NHS Trust involved multiple steps and several actors. Decisions on whether or not to open specific studies rested primarily with the relevant clinical directorate who would

vet the study for its appropriateness, scientific merit and feasibility. Other processes were centralised by the Trust's Research and Development (R&D) governance office, like the sponsorship review (that is, deciding whether the Trust will take responsibility for the study and study compliance) or assisting researchers to gain approvals from national regulatory bodies like the Medicines and Healthcare products Regulatory Agency (MHRA) and the Health Research Authority (HRA). With approvals in place, R&D would then assess whether sufficient resources were available to support the study (the capacity and capability review). Completing this process was often both onerous and time consuming. As a result of the COVID-19 pandemic, substantial parts of this process were reconfigured, as we detail below.

## Methods

### Case

This is a case study of how the embedded research infrastructure at one NHS Trust was repurposed to support the delivery of frontline care and COVID-19 research. The case study method allowed us to track how the research system was adapting in real time, and enabled an in-depth look at the processes and mechanisms that have underpinned operational changes [13]. As an instrumental case study, one that focuses on socially, historically and politically situated issues, we use a single site to examine issues that are also faced by other hospital Trusts [14]. We employed an online questionnaire of research-involved staff, document analysis of emails and official national and Trust documents, observations of planning meetings and semi-structured interviews. Data were collected from individuals working in four levels of the research infrastructure: (1) central research oversight and governance (including R&D leads and research governance staff); (2) principal investigators (PIs); (3) the research delivery workforce (including research nurses, clinical research practitioners, data analysts and research managers); and (4) Patient and Public Involvement (PPI) managers and PPI representatives. Triangulating these four data sources and four levels allowed us to consider the representativeness of our data across the case. Redeployment figures and wider workforce information were provided through a request to SLAT's research management office.

### Sampling and data collection

Data were collected by DW, RFG and HC over a period of six months, from May to October 2020. In the first stage of research, an online questionnaire was disseminated to all research-involved staff at SLAT (approx. 700) on 18th May 2020 via pre-existing mailing lists. The questionnaire closed on 10th June 2020 with 170 responses, yielding a response rate of approximately 24%. Whilst 24% would be an inadequate response rate for statistical analysis [15], it was not intended as a validated survey, but rather a method to gain a broad understanding of staff's experiences of the COVID-19 research response, with most questions open-ended. We received completed questionnaires from nearly a quarter of research staff during the pandemic. The questionnaire also enabled us to identify and recruit a maximum variation sample of staff involved in the research response across the four groups to interview. Interviews allowed us to explore in more depth some of the recurring themes first identified in the questionnaire.

Interview participants were also recruited using purposive and snowball sampling with an aim to maximise the representation of a variety of experiences across the case [16]. Key staff within SLAT were identified based on searching the Trust's website, reviewing staff lists and by speaking to senior personnel for guidance. Interviews were conducted digitally on Microsoft Teams and were recorded and transcribed verbatim. Interviews focused on participants'

work prior to the pandemic, how this work has changed as a result of COVID-19, and the short and long term impacts of COVID-19 on health research more broadly.

Additionally, we obtained permission to observe the regular research prioritisation meetings convened by the Trust's Director of R&D. These meetings took place over Microsoft Teams once or twice a week and were attended by an average of 10 senior clinical, research and research delivery leaders per session. We attended the meetings as non-participant observers, taking notes and recording proceedings. Recordings were transcribed verbatim. We also analysed all documents that were produced or circulated in connection to the prioritisation meetings. These included email discussions about specific projects, national directives, Trust protocols as well as the applications submitted by investigators to the prioritisation committee.

Lastly, we attended the handful of PPI meetings that were held by the few active PPI groups during this period. We participated in discussions about specific research projects and heard participants' experiences of PPI during the pandemic. PPI is a core part of the pre-COVID-19 research and research design process [17]. It was therefore important that changes to PPI were considered within our study. We were also able to present our research and get feedback from groups about our aims. PPI meetings were not recorded, but detailed notes were taken during each session.

Conducting qualitative research during the COVID-19 pandemic has required us to adapt data collection methods to accommodate restrictions on face-to-face meetings and access to the hospital. Studies note that while video conferencing has many benefits, issues such as the familiarity of participants with online platforms and access to technology and high-speed internet can be barriers to the successful use of these technologies in interviewing [18, 19]. We experienced only a handful of technical problems in our interviews. In all but two instances, interviews were conducted with cameras on so that we could observe non-verbal communication [20].

## Analysis

Our data were managed and analysed through NVivo 12 using a two stage process [21]. In the first stage, we analysed the data for a descriptive and narrative account, paying attention to the contours of the emerging response to COVID-19, including national and Trust decision-making and action [22]. In the second stage we used thematic analysis to develop an analytic account based on emerging themes [21, 23]. Data were coded for key themes independently by DW, RFG and HC iteratively throughout the data collection process. Codes and core themes were then discussed and verified across the researchers. As part of our analysis process, we also presented initial findings to research staff at SLAT and at another NHS Trust. These methods of challenging our analysis both internally and externally were crucial for ensuring we reflected on our own influences on the data and the data's utility beyond our specific case [24].

Ethics approval for the study was granted by North East—Newcastle & North Tyneside 2 REC (reference: 20/NE/0138).

## Results

We completed 24 interviews, lasting from 24 to 105 minutes (mean average of 52 minutes), observed approximately 40 hours of research prioritisation meetings and 4.5 hours of PPI meetings, and received 170 responses to the questionnaire. In the results that follow our interview participants are divided into four groups. We identify participants using a letter to denote group and number for interview within this group:—G-n (Governance/R&D staff), R-n (Research leaders/PIs), D-n (Research delivery staff), P-n (PPI managers). 3 participants sit in more than one of these groups due to their multiple roles within the Trust. These participants

**Table 1. Questionnaire participants by role descriptor.**

| Role descriptor | No. |
|---|---|
| Research governance, research management and/or research administration staff | 40 |
| Researchers (clinical academics [including clinician scientists and clinical lecturers, clinical senior lecturers, clinical readers and clinical professors], research academics [including research fellows, senior research fellows, lecturers, senior lecturers, readers and professors], clinicians) | 36 |
| Research Delivery Staff (including Research Nurses, Research Midwives, Research Allied Health Professionals, Clinical Research Practitioners, Laboratory and Support Staff, Project Managers, Trial Coordinators, Data Managers and Research Assistants) | 79 |
| Did not answer | 15 |
| **TOTAL** | **170** |

were interviewed using questions from interview guides for all relevant groups. Questionnaire participants are identified as Q-n, followed by a brief description of their role. See Tables 1 and 2 for a breakdown of participants.

## Centralisation: Prioritising COVID-19 research and redeploying research staff

Centralisation within the research apparatus occurred across two levels.

**National decision-making.** At the outset of the pandemic, DHSC took steps to assert central control over national research priorities in order to coordinate the national response to COVID-19. This included the shut down or partial shutdown of the normal functioning of the research system. A document circulated throughout the NHS on the 13th March 2020, which included information from 25 separate Trusts, announced that elements of the UK's national R&D infrastructure, including the UK Clinical Research Facilities (CRF) and NIHR Clinical Research Network (NIHR CRN) Coordinating Centre were "joining up working to ensure consistency of approach" and that "currently UK NIHR/RC and EU research funding bodies are in the process of selecting research that will be prioritised for approval and delivery across the NHS during the pandemic." On 16th March 2020 a directive from the DHSC and the Chief Medical Officer (CMO) ordered the suspension of all non-COVID-19-related research and the reorientation of research capacity towards the effort to develop COVID-19 treatments and vaccines [6]. Only those studies funded by the NIHR and where "discontinuing them will have significant detrimental effects on the ongoing care of individual participants involved in those studies" were allowed to continue [6]—in short, those studies where research was the standard of care, for example, with experimental cancer treatments. Decisions on which studies met this threshold were decided at the Trust level. Table 3 documents the scale of the pause in the normal research pipeline at SLAT. Participant G-2 saw this DHSC and CMO directive as an effective way to focus research resources:

**Table 2. Interview participants by role descriptor.**

| Role descriptor | No. | No. (acknowledging multiple roles) |
|---|---|---|
| Research governance, research management and/or research administration staff | 5 | 8 (1 Research leader, 2 Research delivery staff) |
| Researcher leaders (Principal investigators) | 7 | 7 |
| Research delivery staff (including Research Nurses, Research Midwives, Research Allied Health Professionals, Clinical Research Practitioners, Laboratory and Support Staff, Project Managers, Trial Coordinators, Data Managers and Research Assistants) | 9 | 9 |
| PPI staff | 3 | 3 |
| **TOTAL** | **24** | **27** |

**Table 3. Status of non-commercial studies.**

| Status | 07 Oct 2019 | 10 Jan 2020 | 14 Apr 2020 | 10 Aug 2020 | 08 Oct 2020 | 07 Jan 2021 |
|---|---|---|---|---|---|---|
| Setup | 215 | 208 | 207 | 239 | 268 | 255 |
| Open | 846 | 860 | 13 | 201 | 586 | 642 |
| Recruitment paused (COVID) | 0 | 0 | 800 | 537 | 104 | 62 |
| Suspended (non-COVID reasons) | 24 | 24 | 23 | 27 | 43 | 58 |
| In follow-up | 156 | 161 | 220 | 269 | 297 | 294 |

> I think the really helpful bit was the sort of diktat from Chris Whitty and Louise Wood at DH [Department of Health and Social Care] to say, "Stop everything that's not COVID." [. . .] So, to actually have something centrally that said, "No, you're not actually allowed to do that because we've got to focus on the COVID stuff," was very helpful because people just stopped asking–which was great. And we were freed up to change processes as we needed to.

Following this directive, a new system of badging certain studies as of Urgent Public Health (UPH) was established, run by DHSC and the CMO. All clinical studies including COVID-19 treatment and vaccine trials that hoped to recruit patients within NHS sites were required to apply for UPH status. An Urgent Public Health Group was convened, chaired by Nick Lemoine, the medical director of the NIHR CRN. The group was responsible for deciding which protocols to label UPH, based on evaluations of scientific merit, feasibility and greatest potential patient benefit [25, 26]. Of the 1600 research protocols received by the CMO from March 2020 to February 2021, only 83 were considered national priorities [5, 27]. Once a study had received UPH badging, hospital sites like SLAT were required to open them, if resources were available.

This centrally-organised prioritisation of COVID-19-related research removed the authority of individual Trusts and directorates to shape their own research portfolios. This was an unprecedented move by the DHSC, but allowed resources to be concentrated on studies deemed to have the greatest potential impact.

**Trust-level decision-making.** In order to enact the DHSC mandate to prioritise COVID-19 research, SLAT created a Trust-level prioritisation process. Twice-weekly prioritisation meetings commenced early April 2020 and were attended by research governance managers, research delivery managers and senior clinicians as well as representatives from the local Clinical Research Network and partner hospitals within the network. The aim of the prioritisation meetings was to protect resources and ensure capacity to undertake UPH-badged research. However, it also ensured effective, timely communication with PIs, helped identify local PIs for new COVID-19 studies led elsewhere, and managed the pause and restart of all non-COVID research. A proforma was introduced to facilitate and standardise prioritisation decision-making. Investigators were asked to provide information summarising their projects, resource requirements and whether they had received UPH badging. Proformas were reviewed during these meetings. By the end of February 2021, this group had reviewed 170 research projects using these proformas across 68 meetings, approving over 80 studies for local setup.

During the first wave of the pandemic, prioritisation group meetings focused mainly on how to open UPH-badged studies, as all other new research had been halted. One important exception was COVID-19 studies that require little or no NHS resource and took place within a single NHS site. These studies were also discussed in these prioritisation group meetings, often with a focus placed on clinical and academic merit. Most of the studies that fitted these

criteria and were approved by the prioritisation group involved university researchers analysing patient data collected and pooled in the COVID patient 'data lake'. This enabled the Trust to maintain research activity in areas not explicitly identified as urgent public health. The research reported in this article was approved through this process.

The joined up approach between national and local decision-making however did cause confusion and frustration. The process of determining whether or not a study would be badged UPH and thus allowed to proceed was initially opaque to Trust researchers and R&D, and the national UPH review process often took weeks from application submission to outcome. Furthermore, the decision to grant a study UPH was and remains out of the hands of the sites that are tasked with delivering this research, even when internally questions were raised about the appropriateness, feasibility or scientific merit of the study. Some researchers designing studies to address key issues in relation to COVID-19 struggled to negotiate the system:

> In terms of national COVID studies, we tried to get a number of studies up and going, focusing on older patients. And ran into quite a lot of obstacles and barriers. [..P]eople weren't certain whether this was research or whether it was quality improvement, audit-type, survey-type work. And that was pretty frustrating, not being able to get clear answers on that from the senior team within R&D. And access to data was very difficult. So, despite lots of conversations about why we really needed to be focusing on older patients, the majority of people with COVID, the biggest impact being in care homes, it was quite frustrating getting hold of people who could actually sign off on studies that we would have like to have done (R-7).

At the Trust level, the prioritisation of research was also important because of the reduction in available research delivery staff. As Table 4 documents, the clinical research delivery workforce, which totalled 165 on 14th April 2020, was reduced by 79% or 131 staff members during the peak of the first wave due to redeployment to frontline care. A further 52 non-clinical research staff were redeployed to support other Trust activity. With such a reduction of staff, the ability to maintain even those studies which had not been halted was not certain and indeed many studies required changes and protocol deviations as a result. A key point of discussion in all prioritisation meetings was the resourcing requirements of proposed studies and how these requirements might be managed alongside existing commitments. In tandem with these discussions, work was done by the research delivery manager to create a central register of research delivery staff within the Trust. The push to centralise oversite of research delivery staff was initially driven by the requirement to rapidly redeploy staff including nurses and clinical trials practitioners to support the Trust's emergency response but it was also crucial to the

**Table 4. Clinical and non-clinical research staff redeployment on 14th April 2020.**

| Clinical research staff redeployed to clinical roles | | | Non-clinical research staff redeployed to non-clinical roles | | |
|---|---|---|---|---|---|
| Role | Destination | No. | Role | Destination | No. |
| Adult Research Nurse | ICU, COVID wards, NHS Nightingale London Hospital | 50 | Non-clinical R&D Staff | Ward clerks | 31 |
| Paediatric Research Nurse | Evelina clinical activity, NHS Nightingale London Hospital | 27 | Project Managers | Tactical sub-groups | 2 |
| Research Midwife | Routine clinics, maternity helpline | 24 | Research Technicians | Viapath | 7 |
| Clinical Research Practitioner | ICU turning team | 14 | Research staff | Data entry | 6 |
| Unassigned | | 16 | Research staff | Bereavement centre | 2 |
| | | | Research staff | Cancer centre outpatient clinics | 4 |
| **TOTAL** | | **131** | **TOTAL** | | **52** |

prioritisation group's understanding of the availability of research resources. Prior to the pandemic, there was no central list of all research delivery staff at the Trust, as D-2 discusses:

> A benefit was actually establishing who all the staff are. The systems we have in R&D which relate to where staff sit within the Trust system depends on where they're funded from. And because research teams have lots of mixed types of funding, some of the staff are visible to me through the systems and some aren't. So, the only way for me to know who all the staff were, was to manually myself, physically ask. There was no system anywhere that listed who the research staff are.

In addition to being redeployed to the clinical frontline, research staff were also pulled from across the Trust's many directorates to form a new dedicated COVID-19 research delivery team. This team became responsible for the rapid set up and roll out of COVID studies of national and international importance, like the Oxford AstraZeneca vaccine trial, among others. Centralising oversight and management of the previously dispersed research delivery workforce enabled SLAT's research system to react quickly and flexibly to the rapidly evolving clinical demands and research requirements of the pandemic.

While research activity was centrally coordinated within SLAT, R&D were initially left out of Trust emergency planning. An organogram produced by the Trust to represent its emergency response plan did not include R&D or any element of the research system, and a briefing document prepared by SLAT R&D for the Trust's Gold Tactical Command Unit dated 14th April 2020 noted this absence, and that there was also no "obvious place in the structure for R&D to naturally sit." Participant G-3 reflected on what was perceived initially as a failure to consider the role of research:

> I think [. . .] the Trust essentially, corporately, hadn't involved the R&D department in what they were thinking. [. . .] We didn't have a tactical subgroup where everybody else, every other area in the Trust had a tactical subgroup. [. . .] There was nothing in place. You know, we've all voiced this, certainly in meetings at the senior management level–is that, and the words used were, "R&D has been forgotten." We were forgotten. So, what the Trust had set up and which is, I think, probably a policy or a set of actions that they have for crisis management [. . .] was very militarily organised. [. . .] And we didn't slot in, nor were we invited on to any of those tactical groups. And didn't have representation on gold or silver command either. So we were left out of that whole process. [. . .] We had to make real efforts to reach out and offer up. We felt that obligation and we did that.

By late April 2020, R&D were fully integrated into the Trust's Gold Tactical Command Unit. By this time, however, the prioritisation process had been implemented and oversight of research delivery staff had been centralised, facilitating redeployment to frontline care and COVID-19 research. While the research system contributed staff and other resources to the Trust's emergency response, it did so at its own initiation.

## Pace of work: Shifting gears for the COVID-19 response

One of the most striking aspects of the research infrastructure's response to the pandemic was the sheer pace of activity and change. The sociological literature on pace suggests that demands for faster productivity are common, and indeed this demand can be seen in the health services literature which often criticises clinical research for not moving fast enough [28–31]. However, the sociological literature also notes the importance of considering where things slow down or

even halt [28, 32]. In this section we document how pace appeared in participants' accounts, acknowledging both areas where there were rapid increases in the speed of research work as well as how research work slowed down in other areas.

**Increasing pace: Redeployment, research set up and research completion.** Particularly within the first wave, it was the "reserve army" (D-3) of the research delivery workforce who were required to act at speed. As per Table 4, staff were quickly released from research duties and redeployed to the frontlines to help deliver care. In addition, all NIHR funded staff with clinical training who were not completing COVID-19 research were asked to prioritise front-line care if their employer asked [6]. Within two weeks, more research delivery staff were redeployed to COVID-19 research teams. Staff were called up one day and told to "come in on the next day" (D-8), and managers were told "they're going tomorrow. This is their last day with you" (D-4).

As pace of redeployment accelerated, so too did the speed of research. The pace with which researchers demanded studies be delivered and set up was "ten times quicker than normal [. . .] as if someone's taken a time warp machine to it" (R-2). Those already working in the research infrastructure were aware that research was vital to the pandemic response and, as one participant (D-1) explained:

> we needed to start the research while we're right in the middle of the surge in numbers. And so [. . .] you have studies that come, they need to be set up tomorrow, recruit the first patient by the end of the week.

Such shifts in normal timeframes for work were facilitated in part through centralisation, as noted above. "The real step change," research manager G-4 suggested, "was having a Prioritisation Group and having [the] team agree a fast-track way of doing things." Alongside streamlined approval and set-up processes, wider research infrastructures and research practices were adapting at great speed:

> I was amazed that, for example, by the end of March, there were–I counted them– 13 granting agencies that, some way or another, had calls on urgent COVID-19 research (R-4).

As a result of these rapid research projects, new knowledge was being produced at an unprecedented rate, as one participant succinctly put it, "science doesn't usually change that quickly" (D-9). This speed was met with enthusiasm by PIs and research delivery staff alike, but also caused some nervousness. Some were concerned, for example, that PPI had "dropped off the radar" (G-3), whilst others were wary of publication prior to peer review:

> the [. . .] thing which is a challenge is that we're pre-printing research, we're putting preprints out when we're submitting to journals, because–and we're rushing to get the preprints out. [. . .] And I guess that's good. But it is also a bit of a–a stresser because [. . .] maybe we haven't quite got the message right yet (R-1).

Others warned that the pace of research during the first wave of the pandemic came at a human cost. Some researchers had vastly increased workloads, "going at max [. . .] for 5 months" (R-1), where in some cases "there's not been a single day when [they've] not been working in the laboratory including all Sundays and Saturdays, Easter and so on" (R-4). Whilst some enjoyed this fast-paced moment, for those closer to the frontline it has caused anxiety. As one participant (G-5) explained, "we've been fire-fighting", and at least one member of staff, another explained, "can't come near the hospital. She has panic attacks" (D-3). Whilst it

has already been documented that critical care staff's mental health has suffered in the pandemic, these participants suggest there may also be concern for the staff involved in the research response [33].

Seeing what is possible within the exceptional circumstances of a global pandemic led some researchers and PPI managers to question the normal slower pace of regulatory approvals and assert, "if you can do it during COVID-19, you can do it any other time" (R-6). The often slow processes such as ethical approvals, data sharing guidelines, funding applications, and study set-up was a common comparator to what has been possible during the COVID-19 pandemic. Yet, as G-1 explained: "The reason [research processes have] been quicker is just because there's been less studies." This is evident in SLAT's own R&D data. Table 5 documents the difference in study numbers and timeframes from initial sponsorship review to final capacity and capability approval (allowing the site set up and recruitment to commence) across 3 financial years. While some approval processes were adapted, generally research governance requirements, both internal to the Trust and at the regulators the MHRA and the Health Research Authority, remained the same. The quick approval processes were possible because no new non-COVID-19 studies were reviewed, COVID-19 studies were processed as quickly as possible and almost all non-COVID-19 related research was halted.

**Slowing or halting non-COVID-19 research.** For some investigators, the halting of non-COVID-19 research led to a slower pace where researchers could play catch up. "People have just been writing up their papers" (R-3), and this period of time "gave [. . .] the opportunity, freed up time" (R-6) to apply for grants. Whilst many tried to set up studies so they were ready to go when restrictions were lifted, they also found that "regulatory bodies have been slower" (R-6) due to their focus on COVID-19. It was apparent that these researchers had more time to engage in PPI whilst putting these grants together–one PPI manager working in cancer (P-3) suggested "PPI activity has probably increased" during the pandemic. Whilst many researchers were understanding of the need to halt research, others found it devastating for patients and the reputation of UK research. These researchers (R-3 and R-6) pointed to other international contexts where they saw standard research continuing. Researcher R-6 was surprised "with the UK being such a [. . .] clinical trials powerhouse", that decision-makers didn't "do everything it could to retain that reputation even through the COVID-19 crisis."

On 21st May 2020 the DHSC and NIHR circulated a framework for restarting new and paused non-COVID-19 research. Stratifying research studies into three levels of priority, this framework made no distinction between commercial and non-commercial research. Using this framework, the Trust implemented its operational Restart Plan the week commencing 1st June 2020. Recommendations on which research studies were important or urgent to restart within each directorate was managed a directorate level, with the Prioritisation Group acting as the Trust-level decision making body for the restart plan. The Prioritisation Group continued to meet weekly to approve restart plans for research projects. By mid-summer restart was well underway but the pace of resuming all these studies could not match the pace that research stopped, and researchers were concerned that they "haven't really been able to pick up our trial recruitment in between [waves], because recovery has been so slow" (R-5). The

**Table 5. Time from sponsorship to issuing capacity and capability approvals (including only non-commercial studies required to complete the full R&D review process from sponsorship through to capacity and capability approval).**

| Financial Year | Sponsorship reviews started | Capacity and Capability approvals issued | Mean timeline | Median timeline |
|---|---|---|---|---|
| 2018/19 | 129 | 95 | 206 days | 189 days |
| 2019/20 | 138 | 66 | 224 days | 187 days |
| 2020/21 | 66 | 23 | 65 days | 62 days |

time of "let's get back to normal quickly because COVID's over", participant R-2 explained soon turned to "actually, let's not rush back into things because we don't know what's coming." At this point the centralisation of research infrastructures hindered speed rather than aided it–one research governance manager (G-4) suggested that "we need to respect the decision-making of the research managers and matron and the R&D leads now", but instead studies were "number 507 in the queue", and having to "wait another week for this prioritisation meeting" whilst "people are really scared about their finances [. . .] frightened about not finishing [. . .] patients are waiting."

## Adopting new and virtual working practices

The response to COVID-19 pandemic has resulted in broad shifts in working patterns across the labour market, and will likely lead to longer term transformations to work practices stemming from these temporary changes [34–36]. In health, research highlights the accelerated adoption of digital and virtual working practices as a result of COVID-19, such as the use of telemedicine in secondary care [37–39]. The implementation of new working practices, taking advantage of digital technologies for communication and the adaptation of existing processes so that they can be completed (at least in part) during the pandemic are also crucial elements of the research response to COVID-19, particularly for facilitating the continuation of research.

**Reducing patient visits.**   Clinical research is a highly regulated domain, with strict oversight on practices and procedures, and reporting requirements overseen by multiple regulators. While research setup and governance processes became more centralised, the successful conduct of research during the pandemic required a degree of flexibility and creative adaptation. The move to more remote or virtual ways of completing, supporting, regulating, and facilitating research relied on the speedy adoption of new technologies and ways of working.

On 12th March 2020, the MHRA issued guidance to sites and investigators "regarding protocol compliance during exceptional circumstances" [40]. The guidance stated that the MHRA recognised "the difficult current situation" and advised on how to manage trials during the pandemic [40]. The MHRA also noted in this guidance and on the MHRA Inspectorate website that a redistribution of human resources during the pandemic:

> may mean certain oversight duties, such as monitoring and quality assurance activities might need to be reassessed and alternative proportionate mechanisms of oversight introduced (such as phone calls, video calls) to ensure ongoing subject safety and well-being. We would advise a brief risk assessment and documentation of the impact of this [40].

While this guidance came before the formal research shutdown, it remained important, especially for the small amount of research which was allowed to continue because it was the best or only treatment option left available for patients. However, research practices and trial protocols needed to be adapted, particularly as there were restrictions on who could physically visit hospital sites, as G-5 highlights:

> If a protocol says that a participant will have a visit at week 1, week 2, week 3 and week 4 and those are protocol visits–it's unacceptable not to do those visits. They are protocol deviations. However, during the real surge of the pandemic, those visits couldn't be done. They couldn't come in and have an MRI scan, and ECG and bloods taken. What they did have was someone contacting them by telephone or by Skype or other formats, media format–to say, "How are you doing? Are you okay? Is there anything you need to report? Keep in touch" (G-5).

Through delaying or adapting follow-up appointment requirements so they could be completed over the telephone or through videoconferencing, many studies were able to maintain some level of continuity. For these research participating patients, other parts of the research process needed sensitive negotiation, as one PI explains in relation to changes in the format of patient consultations:

Some [participants] were actually a bit reluctant and felt a bit fobbed off to be called at home [when] they were due a face-to-face consultation. We had to be a bit careful about that, particularly if we were discontinuing treatment or discharging people from our care. That almost always went badly if we tried to do it remotely. And if we were having a really definitive conversation like that, it was worth–we found, in the end, patients coming up. Other patients were reluctant to come and readily accepted our advice that rather than coming for a CT scan, we just do a chest x-ray when we next saw them. So, there is a difference of approach, which is personal–not particular to their circumstance (R-5).

Balancing the need for face-to-face consultations and the protection offered by telephone or video consultations required thoughtful, individualised decision-making. For other studies however, digital consultation was simply not possible, which lead to investment in supporting people to attend the hospital:

A few studies have been done remotely, but the one that I have taken on, patients really have to come in. So, we had to do a lot of logistic development there, like bringing them in by car, paying for whatever is necessary just to make sure that they continue coming in (D-6).

**Working from home.** Another crucial step in facilitating research and frontline care was asking large numbers of staff to complete their work from home. For some participants, working from home lead to greater productivity, but for many others it meant the blurring of home and work lives. Numerous factors impacted on participants' experiences, from juggling work alongside home schooling and caring responsibilities, to feelings of isolation, through to more practical issues, such as having a space to work at home, having sufficient internet bandwidth and having stable access to Trust systems (see Box 1).

While research staff were transitioning to working from home, research spaces were transformed to facilitate frontline care. By April, two of the four Clinical Research Facilities (CRFs) in the Trust were repurposed to deliver frontline care and training space for frontline staff. The remaining two CRFs were refocused on supporting COVID-19 research. The vacant R&D department's office spaces were also used by Trust staff to facilitate socially-distanced meetings and computer work for those who needed to be onsite. Careful repurposing of offices and clinical space provided the Trust with additional, flexible physical space to assist in the emergency response to the pandemic.

**Digitalising research processes.** Research work still occurred within the normal parameters of how health research is conducted in the NHS. These practices were, however, done differently to adapt to COVID-19 social distancing measures.

Firstly, researchers initially had to find a workaround for consent to research in COVID-19 wards. Because of infection control protocols no materials, including paper consent forms, could be removed from COVID positive wards. As there were no protocols in place to gain consent digitally, staff developed a local workaround, as D-1 explains:

we managed to get some [. . .] work phones so that we could take a picture of the consent [form]. So, the consent [form] was held up to the window [in the COVID ward], the team

**Box 1. Indicative questionnaire responses to: What, if any, challenges have you had to face working from home?.**

*Increased productivity*

I am fortunate enough to have space, a home office and garden space. The provision of IT has made sure that I have been able to maintain and increase productivity (Q-48, *Research governance*)

None, I have actually found it to be extremely productive working from home. I am able to give more support and work closer with my team due to everyone being a video call away. *(Q-41, Research governance)*

*Access to systems, technology and documents*

Not all features of the clinical documentation system IMS were accessible on my private laptop, therefore I still had to come to Guy's Hospital sometimes to fulfil my role—even I belong to the vulnerable group in regard to COVID-19. IT-services tried to help me, but I would have needed a Trust-Laptop which wasn't available. *(Q-6, Research delivery workforce, Research nurse)*

Having to use my own laptop. Remote desktop is very cumbersome, and things take longer to do. Sore back from poor posture and lack of a suitable chair. *(Q-59, Research governance)*

Internet connections bandwidth cuts out or slow as using home personal network and or personal mobile telephone with patchy reception, teleconference and video chat problems, lack of informal chats with colleague and ease of keeping up to date on broader issues from regular face to face meeting *(Q-47, Research governance)*.

To access patient source documents and little patient contact roles *(Q-6, Research delivery workforce, Research nurse)*

*Balancing home and work lives, including home schooling*

Sometimes I feel that I have been working more hours than I would normally do, I feel productive but also feel I need "space" away from work at home. (Q-93, *Research delivery workforce, Clinical research practitioner*)

I have found it very challenging to stay focused on work while family life plays out around me. Also having the children home schooled raises more concerns and increased oversight *(Q-107, Researcher)*

Managing childcare and the physical environment. *(Q-140, Research governance)*

Cross covering other teams work (those who have been redeployed) and ensuring that the standard the carried out is maintained—Maintaining contact with research teams and PI's when many of them have been redeployed or refocusing their attention of COVID-19 studies. *(Q-14, Research governance)*

*Physical co-presence*

The "corridor" conversations that happen in an office / lab-based environment don't happen. These are vital conversations where quick decisions can be made without the setup of meetings and coordinating timing. Decisions therefore take longer, and other outputs and deliverables are inevitably impacted with the time taken up needing to schedule calls, allow for others availability etc. *(Q-149, Research governance)*

Solid computer time. Previously, working from home included a range of computer and paper activities and time for thinking. Currently, I am using computer for remote access almost all my working hours. That is more tiring. Some programmes are slower or interrupted, probably due to challenges with local internet access/demand. I am very comfortable with lone working but do at times miss the companionship of working in the same physical space as others *(Q-164, Research delivery workforce, Research nurse)*

outside could take a picture of the consent form and send it directly through on the Pando app, because [Pando] could have patient details. So, it could then be turned into a PDF and printed and put in the patient file.

Another example of a slow but necessary digital solution was with site monitoring. Site monitoring allows commercial companies and other trial sponsors to visit research sites to assess the quality of the data and ensure study protocols are being followed. Despite MHRA instruction that this "should not add extra burden to trial sites" [40] and that monitors could not be justified as an extra body in the building, these activities are crucial not just for validating data but for hospitals to be able to bill sponsors for the completed research. Workarounds were further limited because of data protection regulations that prevent the digital transfer of

patient data or remote access to Trust systems by external individuals. Where site monitors would usually work alone on site, it became a long and arduous process:

> a member of the research team within the Trust sits at a screen and shared that screen through Microsoft Teams with the external person. So, no data is held, no recordings are being done, no data is transferred. But it's very, very labour-intensive. (G-5)

Whilst workarounds were quickly found for some research practices, others took longer. Despite the fact that Patient and Public Involvement in research (PPI) is a core element of contemporary UK health research [17], there was initially "zero PPI" (G-1). Rather PPI group managers focused on care work: "putting them in touch with local services that could do things like pick up prescriptions for them, get shopping, get the food boxes delivered" (P-1). It was only with time that not only did researchers planning non-COVID research begin to engage more than usual with their PPI groups, but that funders and regulators demanded that PPI should still be prioritised even in emergency research [41, 42].

While researchers voiced concerns about the equity of shifting online and assumptions about who will and will not engage with online PPI, this did not appear to be a problem in practice:

> There's often a sort of an ageism about who can–it's like kind of what you were just saying about older people can't do PPI. Well, bollocks. I mean actually they've been as responsive to this pandemic as anybody else. The rates of use of, you know, technology, has like sky-rocketed in the over 65s, because of their need to talk to their grandchildren etc. So, you know, they are adaptive (R-1).

R-1's experience was echoed by PPI representatives. Reflecting on the move online, these representatives noted some disadvantages, such as the absence of many social aspects of attending PPI meetings, and video fatigue. But participants were generally positive about the potential of virtual PPI for involving those who cannot always travel long distances due to their illnesses, those who work full-time but could attend an hour session online in their lunch break, and representatives in different countries.

In short, the process of realigning and digitalising research practices was not simply one that sped up research and productivity, but it involved a set of necessary, labour-intensive workarounds. It did, however, also bring about possibilities for long term positive effects, such as diversifying involvement in PPI groups.

## Discussion

COVID-19 has brought to the fore the critical importance of the UK's clinical research infrastructure which has over the past 15 years become increasingly embedded within the NHS. It has enabled NHS hospitals to deliver research of global importance at an unprecedented pace while simultaneously providing critical care for record numbers of acutely ill patients. We provide an analysis of how this was possible through an in-depth case study of the transformations and reconfigurations of the research system at one research-intensive Trust. Our data show that a large-scale reorganisation of research staff, research infrastructures and research priorities took place during the first few weeks and months of the pandemic. We have documented many of the changes in organisational structure, national policy and everyday working practices that facilitated the Trust's response to COVID-19. These rapid changes have brought about new ways of working, and new perspectives on the role of research which may have far reaching consequences for the future of the clinical research system in the UK.

The pandemic occasioned a large-scale mobilisation of research staff as a "reserve army." Research staff were crucial in supporting the care-function of NHS hospitals during the first wave of the pandemic. At the same time, the embedded research system helped to streamline, facilitate and deliver rapid COVID-19 research.

Our study documented some of the challenges that the research system has faced in seeking to operate in a COVID-safe manner. At the same time, our participants described instances of improvisation in order to adapt protocols to the COVID-19 environment. Research staff developed effective practical solutions borne out of necessity, rather than the result of prior planning. This points to the resourcefulness of research staff, but also highlights the ways in which the research system was initially largely absent from existing emergency planning within the health system.

Our research was conducted while the Trust we were studying enacted national COVID-19 policy, responded to local care needs and supported clinical research during a global pandemic. This allowed us to observe these events unfolding while gathering data in a COVID-safe manner. But the pandemic created limitations as well, especially impacting the range of methods we were able to use. While working digitally did give us a first-hand experience of how a large proportion of the decision-making infrastructure had to move online, it limited our access to frontline care and everyday research activity.

There are also limitations of looking at a research active Trust like SLAT. While research is increasingly becoming a routine component of all NHS settings, SLATs size and existing research portfolio meant there was a large amount of resource available to redeploy towards COVID-19 care and research delivery. This picture may not be representative of all NHS Trusts, particularly those that are smaller, where less research takes place. Such resource, particularly in the form of biomedical research infrastructures embedded within NHS Trusts, have provided what Roope et al. label 'option value' in research, additional capacity to support public good, which in normal times may appear an inefficient use of resource [43]. Roope et al. highlight that, in comparison to funded, individual research studies, funding research infrastructures allows greater flexibility and speed of response when emergencies arise, such as the COVID-19 pandemic. While the research workforce, funds and infrastructures were used to support other research prior to COVID-19 (as opposed to being excess capacity), the ability of such resource to be reallocated to COVID-19 at such pace underpinned much of the UK's success in its research response and much of the work described in this paper. It is important to acknowledge, however, that research capacity is distributed unevenly throughout the NHS, and resources such as Clinical Research Facilities and Biomedical Research Centres tend to be situated in major teaching hospitals and trauma centres rather than geographically more localised hospitals. More research is needed to understand how this unequal distribution of resources affected outcomes of care and research during the pandemic.

In documenting how the pace of research work changed dramatically during the pandemic, both in terms of increasing the speed of certain activities and decreasing the speed of others, our paper also contributes to broader discussions of pace in clinical research. In particular, the key question—how do we most effectively streamline the research pipeline, from bench to bedside? Hanney et al. highlight the potential to overlap parts of the translational research pathway to speed up the process, and some of the barriers to this, such as ethical approvals and resourcing issues [30, 31]. Many of these issues were removed during the pandemic because of the targeting of resources towards COVID-19 research. On a more practical level, however, our analysis suggests some ways that the research system may be adapted in the future. The potential offered by digital communications to facilitate certain research and PPI activities have led some clinical researchers to question the necessity for research participants and patients to always attend hospital sites for consultations. Trust-level research prioritisation has proved

positive in managing finite local resources as effectively as possible, enabling a more holistic view of the research portfolio at a local level as well as take into account national priorities. At the same time, it is clear that the new technologies and new ways of working that were developed to cope with the crisis are not automatically more efficient, and there is a danger that some key steps such as adequate PPI might be overlooked when research pace is increased. Further research and planning will be needed to develop suitable governance processes to facilitate research activities both when on a crisis footing, and in more routine practice. Wider investment in networked digital applications and hardware (such as Trust compliant laptop computers) is needed to facilitate better working from home.

Our study suggests a number of additional lessons for future national emergency planning and policy. Research infrastructure must be better included in advanced planning, both in terms of the personnel, equipment and other resources that can be made available for redeployment as well as the direct impact that research can make. The capacity to develop new treatments and vaccines should be treated as a strategic asset that is a central part of any emergency response. This has been recognised at the national level, and internationally [1–3], but our data suggest that it has not fully translated into Trust-level operations. Planning for future emergencies should include protocols for the rapid establishment of strategic research prioritisation and redeployment of research infrastructure and capacity. Our data also show that throughout the pandemic, there remained a demand for public input in research, which should be included in future emergency planning. Public input is vital in clinical research, especially in an emergency response which requires publics to respond to clinical-expert advice, and planners should recognise it as such.

Future emergency planning must, however, take into account the exhaustion and stress faced by research staff who suddenly found themselves on the front line of a national mobilisation. Research staff experienced the same well-documented stresses experienced by other NHS workers [33, 44]. Emergency planning should acknowledge this human cost and find ways to mitigate such costs and provide support for staff as a national priority.

At a global level, the UK response and its specific organisation, as described within this case study Trust, demonstrates some of the benefits of embedding research infrastructures within a national health provider, and how this set up not only enabled a coherent national response, but also provided staff resource to facilitate such research at great speed as well as support the delivery of frontline care. As we look to the future, how we integrate healthcare and research at more national and global levels are important areas for further research and discussion.

## Acknowledgments

We are grateful to Christopher McKevitt and Nina Fudge for providing astute comments on drafts of this paper and to our participants who shared their experiences and time with us during this period of unprecedented strain on the NHS.

## Author Contributions

**Conceptualization:** David Wyatt, Rachel Faulkner-Gurstein, Charles D. A. Wolfe.

**Data curation:** David Wyatt, Rachel Faulkner-Gurstein, Hannah Cowan.

**Formal analysis:** David Wyatt, Rachel Faulkner-Gurstein, Hannah Cowan.

**Funding acquisition:** Charles D. A. Wolfe.

**Investigation:** David Wyatt, Rachel Faulkner-Gurstein, Hannah Cowan.

**Methodology:** David Wyatt, Rachel Faulkner-Gurstein, Hannah Cowan.

**Project administration:** David Wyatt, Rachel Faulkner-Gurstein, Hannah Cowan.

**Resources:** Hannah Cowan.

**Validation:** David Wyatt, Rachel Faulkner-Gurstein, Hannah Cowan.

**Writing – original draft:** David Wyatt, Rachel Faulkner-Gurstein, Hannah Cowan.

**Writing – review & editing:** David Wyatt, Rachel Faulkner-Gurstein, Hannah Cowan, Charles D. A. Wolfe.

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
