## [Decision Letter · Decision Letter 0]

14 Jul 2021

PONE-D-21-12411

Impacts of COVID-19 on clinical research in the UK: a multi-method qualitative case study

PLOS ONE

Dear Dr. Wyatt,

Thank you for submitting your manuscript to PLOS ONE. After careful consideration, we feel that it has merit but does not fully meet PLOS ONE’s publication criteria as it currently stands. Therefore, we invite you to submit a revised version of the manuscript that addresses the points raised during the review process.

The reviewers noted the interest and importance of your research question and strength of your methods, particularly the multiple forms of data collected and triangulated under pandemic conditions. The reviewers have suggested some minor revisions, which I ask that you please address. For readability and an international audience, I would also suggest removing all but the most common acronyms (COVID, NHS, MRI, CT scan etc) especially for organisations or processes and to read through with an eye to things that might need a bit of explanation/context (e.g. the Pando app). I look forward to receiving your revised manuscript.

We look forward to receiving your revised manuscript.

Kind regards,

Quinn Grundy, PhD, RN

Academic Editor

PLOS ONE

Journal Requirements:

Reviewers' comments:

Reviewer's Responses to Questions

**Comments to the Author**

1. Is the manuscript technically sound, and do the data support the conclusions?

Reviewer #1: Yes

Reviewer #2: Yes

Reviewer #3: Yes

2. Has the statistical analysis been performed appropriately and rigorously? 

Reviewer #1: N/A

Reviewer #2: N/A

Reviewer #3: N/A

3. Have the authors made all data underlying the findings in their manuscript fully available?

Reviewer #1: Yes

Reviewer #2: No

Reviewer #3: No

4. Is the manuscript presented in an intelligible fashion and written in standard English?

Reviewer #1: Yes

Reviewer #2: Yes

Reviewer #3: Yes

5. Review Comments to the Author

Reviewer #1: This is a well undertaken and fortuitous study that explored the impact of the COVID-19 pandemic on clinical research activities in a UK NHS hospital/trust. The research started very soon after the UK entered their First National Lockdown and does well to capture the immediate impact of the pandemic on clinical research. This case study documents the processes followed at one NHS clinical research site and should provide guidance for further research to explore good practices during this pandemic at a National and International level; as well as preparing for future global public health emergencies.

The case study is detailed, well written and explores a range of factors. It is a case study and as such there are no concerns with the methods used. The discussion is in keeping with their findings. They do highlight in the introduction that SLAT is a research active trust. This issue could be further elaborated in their discussion; especially in relation to adding caution to their experiences. Less well-resourced healthcare settings where research was being undertaken prior to the Pandemic may not have managed as well; or may not have been able to support COVID related research as well. The pandemic has highlighted some of the flaws in centralising research activities and resources to a few, larger healthcare settings.

A few very minor issues:

1) Results (Lines 2004-2007): could the authors review the Directive from the DHSC and CMO. The guidance also states: “However, clinical trials or other research studies which are funded or supported by NIHR should continue if discontinuing them will have significant detrimental effects on the ongoing care of individual participants involved in those studies”.

2) Could the authors clarify the data shown in Table 4? It was unclear where the number of 165 clinical research delivery workforce comes from. Is this based on the participants who completed the questionnaire (n=155)? It was also unclear where the number (52) of non-clinical research staff comes from.

Reviewer #2: The research team deserves congratulations for rapidly producing this generally well-written and well-conducted, multi-method, qualitative study that sheds important new light on how the UK health research system mounted what is generally seen as globally the most effective research response to the COVID-19 pandemic. (I believe the explanation of why there are some restrictions on the availability of data seems entirely appropriate in the circumstances).

Particular strengths of the study include the way in which the multi-methods were used in this case study to create a detailed analysis of the various phases of the impacts of COVID-19 on clinical research, and to analyse the impacts on both COVID-19 and non-COVID-19 research.

There is some recent COVID-19 related literature from the health research systems field that might further assist the authors in drawing their conclusions. This point is described, along with a few others, in the numbered comments below which are presented in the order in which they first arise in the text.

1. Line 58: There are obviously some complexities around whether the term UK or England and Wales should be used. In most places in the text, including here, the term UK is used, but on line 198 reference is made to "the research system in England and Wales": perhaps the authors should consider whether it might be useful to add a footnote to explain how the terms are being used in this article?

2. Lines 62/3 and 84: In relation to the "83 nationally prioritised COVID-19 research projects", first, it might be better to state the full period over which they commenced, rather than just "Since January 2020", and second it might be helpful to clarify here whether this category is the same as the studies "classed as Urgent Public Health studies" described on line 84.

3. Lines 80/82: Perhaps the introduction of the abbreviation "SLAT" could be set out slightly more clearly because the first mention, "we provide a detailed exploration of how the embedded research infrastructure in one South London Acute Trust (SLAT)", seems to imply that SLAT refers to a category of trusts, but then on line 82 and elsewhere throughout the text it is clear, of course, that the abbreviation SLAT is being used for a single trust.

4. Lines 335-341, 440, and Discussion: In relation to questions about the speed in which research is produced, reference is made to "The sociological literature", but it might also be useful to refer to the literature that adopts more of a health research systems perspective and analyses how some research has been conducted much more rapidly than usual during the COVID-19 crisis. Some of the analysis includes, for example, discussion of how the usual queues for decisions and resources that cause sometimes seemingly inexplicable delays in research that turns out to have been of considerable importance, were somewhat eliminated by the increased concentration on COVID-19 vaccine research and the increased resources available; see, for example: Hanney, S.R., Wooding, S., Sussex, J. Grant J. From COVID-19 research to vaccine application: why might it take 17 months not 17 years and what are the wider lessons?. Health Res Policy Sys 18, 61 (2020). https://doi.org/10.1186/s12961-020-00571-3

Reviewer #3: Dear Authors. Thanks for this important study focusing on the perspectives of the researchers working with COVID-19. This is an important case giving valuable insights into future challenges. It gives important insights into workload, different ways of working and also preprints that became an issue when media picked up unverified data and never followed up on the studies once peer reviewed. It would be interesting to have your results compared to other countries. Do you know if any similar research have been done that your case can be compared to? I have only a few comments in the attachment.

6. PLOS authors have the option to publish the peer review history of their article (what does this mean?). If published, this will include your full peer review and any attached files.

Reviewer #1: No

Reviewer #2: No

Reviewer #3: No

---

## [Author Response · Author response to Decision Letter 0]

13 Aug 2021

Dear Editors

We would like to thank the Editor and Reviewers for their time in reviewing our manuscript and their very thoughtful and constructive recommendations, which have helped us strengthen our paper. We very much appreciate this opportunity to revise our manuscript. The document below provides a point-by-point summary of how we have responded to the Editor’s and Reviewers’ comments.

Thank you once again for the opportunity to revise our paper, and we look forward to hearing from you in due course.

With very best wishes

David Wyatt 

(on behalf of all authors)

Editor Comments:

Editor’s comment 1 

Authors’ response:

We have reviewed the reference list, which is formatted using the PLOS Endnote style. We have added references 30 and 31 as a result of the reviewers’ comments and updated reference 5.

Editor’s comment 2

1. Please ensure that your manuscript meets PLOS ONE’s style requirements, including those for file naming. The PLOS ONE style templates can be found at

Authors’ response:

We have now reformatted the manuscript in line with these style requirements.

Editor’s comment 3

2. In your Data Availability statement, you have not specified where the minimal data set underlying the results described in your manuscript can be found. PLOS defines a study’s minimal data set as the underlying data used to reach the conclusions drawn in the manuscript and any additional data required to replicate the reported study findings in their entirety. All PLOS journals require that the minimal data set be made fully available. For more information about our data policy, please see http://journals.plos.org/plosone/s/data-availability.

Important: If there are ethical or legal restrictions to sharing your data publicly, please explain these restrictions in detail. Please see our guidelines for more information on what we consider unacceptable restrictions to publicly sharing data:

http://journals.plos.org/plosone/s/data-availability#loc-unacceptable-data-access-restrictions. Note that it is not acceptable for the authors to be the sole named individuals responsible for ensuring data access.

Authors’ response:

This paper draws on a qualitative research study. As per your guidance for qualitative studies (http://journals.plos.org/plosone/s/data-availability), we have made excerpts of the data available within the body of the article and within Box 1. We believe this meets your data availability requirements and counts as the minimal data set.

Editor’s comment 4

Authors’ response:

We have moved our ethics statement to the Methods section. (Lines 177-178) 

Reviewer 1

Reviewer 1, comment 1

This is a well undertaken and fortuitous study that explored the impact of the COVID-19 pandemic on clinical research activities in a UK NHS hospital/trust. The research started very soon after the UK entered their First National Lockdown and does well to capture the immediate impact of the pandemic on clinical research. This case study documents the processes followed at one NHS clinical research site and should provide guidance for further research to explore good practices during this pandemic at a National and International level; as well as preparing for future global public health emergencies.

 

Authors’ response:

Many thanks for taking the time to review our manuscript and for your valuable feedback. We believe the manuscript to be much improved following your input.

We have added sentences to the Discussion to highlight where further research is needed, particularly in hospital settings where there are not vast research resources that can be redeployed to support care and COVID research. We have also warned about the importance of public input. (Lines 653-654, 686-688)

Reviewer 1, comment 2

The case study is detailed, well written and explores a range of factors. It is a case study and as such there are no concerns with the methods used. The discussion is in keeping with their findings. They do highlight in the introduction that SLAT is a research active trust. This issue could be further elaborated in their discussion; especially in relation to adding caution to their experiences. Less well-resourced healthcare settings where research was being undertaken prior to the Pandemic may not have managed as well; or may not have been able to support COVID related research as well. The pandemic has highlighted some of the flaws in centralising research activities and resources to a few, larger healthcare settings.

Authors’ response:

We have now added an acknowledgement of the potential differences in research and research resourcing across NHS Trusts to the Discussion section. While we can’t comment specifically on how those in NHS Trusts other than SLAT experienced COVID-19 and adapted to deliver care and COVID-19 research, we now acknowledge the value of embedded research infrastructures in facilitating flexibility and pace in emergency situations, which may not be reflected across all NHS Trusts. (Lines 637-663)

Reviewer 1, comment 3

1) Results (Lines 2004-2007): could the authors review the Directive from the DHSC and CMO. The guidance also states: “However, clinical trials or other research studies which are funded or supported by NIHR should continue if discontinuing them will have significant detrimental effects on the ongoing care of individual participants involved in those studies”.

Authors’ response:

Thank you for this point. We have now updated the manuscript to acknowledge these aspects, highlighting that research could continue when it was the standard of care for patients. (Lines 211-218)

Reviewer 1, comment 4

2) Could the authors clarify the data shown in Table 4? It was unclear where the number of 165 clinical research delivery workforce comes from. Is this based on the participants who completed the questionnaire (n=155)? It was also unclear where the number (52) of non-clinical research staff comes from.

 

Authors’ response:

Thank you for pointing out this potential confusion. We have now relabelled the column titles of Table 4 to stress the two different workforces (those that are clinical (131) and those that are non-clinical (52) that were redeployed) and made it clear in text that 165 is the total number of clinical research staff as at 14 April 2020 (and not the number that participated in our study). (Line 291/Table 4, line 295)

Reviewer 2

Reviewer 2, comment 1

The research team deserves congratulations for rapidly producing this generally well-written and well-conducted, multi-method, qualitative study that sheds important new light on how the UK health research system mounted what is generally seen as globally the most effective research response to the COVID-19 pandemic. (I believe the explanation of why there are some restrictions on the availability of data seems entirely appropriate in the circumstances).

Particular strengths of the study include the way in which the multi-methods were used in this case study to create a detailed analysis of the various phases of the impacts of COVID-19 on clinical research, and to analyse the impacts on both COVID-19 and non-COVID-19 research.

There is some recent COVID-19 related literature from the health research systems field that might further assist the authors in drawing their conclusions. This point is described, along with a few others, in the numbered comments below which are presented in the order in which they first arise in the text.

Authors’ response:

Many thanks for your thorough review of our manuscript. We have addressed all of your comments and believe the text to be much improved. Please find point by point response to your comments below. 

Reviewer 2, comment 2

1. Line 58: There are obviously some complexities around whether the term UK or England and Wales should be used. In most places in the text, including here, the term UK is used, but on line 198 reference is made to “the research system in England and Wales”: perhaps the authors should consider whether it might be useful to add a footnote to explain how the terms are being used in this article?

Authors’ response:

Thank you for pointing out this complexity. We have now removed England and Wales from line 258. While there is a difference between the devolved nations of the UK, the pausing of research occurred across the UK, so removing reference to England and Wales avoids overcomplicating the point. (deletion on line 205)

Reviewer 2, comment 3

2. Lines 62/3 and 84: In relation to the “83 nationally prioritised COVID-19 research projects”, first, it might be better to state the full period over which they commenced, rather than just “Since January 2020”, and second it might be helpful to clarify here whether this category is the same as the studies “classed as Urgent Public Health studies” described on line 84.

Authors’ response:

Thank you for this comment. We have amended the text to highlight that these are Urgent Public Health (UPH) studies and removed “Since January 2020” to avoid any confusion. There were no UPH studies before this date. We have also updated the number of studies from 83 to 95 (as further studies have received UPH badging since this reference was added to the manuscript). (Lines 61-62)

Reviewer 2, comment 4

 3. Lines 80/82: Perhaps the introduction of the abbreviation “SLAT” could be set out slightly more clearly because the first mention, “we provide a detailed exploration of how the embedded research infrastructure in one South London Acute Trust (SLAT)”, seems to imply that SLAT refers to a category of trusts, but then on line 82 and elsewhere throughout the text it is clear, of course, that the abbreviation SLAT is being used for a single trust.

Authors’ response:

We have updated the text to make it clear that SLAT is a pseudonym. (Lines 81-82)

Reviewer 2, comment 5

4. Lines 335-341, 440, and Discussion: In relation to questions about the speed in which research is produced, reference is made to “The sociological literature”, but it might also be useful to refer to the literature that adopts more of a health research systems perspective and analyses how some research has been conducted much more rapidly than usual during the COVID-19 crisis. Some of the analysis includes, for example, discussion of how the usual queues for decisions and resources that cause sometimes seemingly inexplicable delays in research that turns out to have been of considerable importance, were somewhat eliminated by the increased concentration on COVID-19 vaccine research and the increased resources available; see, for example: Hanney, S.R., Wooding, S., Sussex, J. Grant J. From COVID-19 research to vaccine application: why might it take 17 months not 17 years and what are the wider lessons?. Health Res Policy Sys 18, 61 (2020). https://doi.org/10.1186/s12961-020-00571-3

Authors’ response:

Thank you for raising this important point. We have referenced this text and the health services literature within the initial introduction to the pace section. To our Discussion section, we have also now added a text on pace that goes beyond the everyday practices in the hospital, drawing on Hanney et al’s work (as you kindly suggest) and that of Roope et al. (on option value). (Lines 352-353 and 656-663)

Reviewer 3

Reviewer 3, Comment 1

Dear Authors. Thanks for this important study focusing on the perspectives of the researchers working with COVID-19. This is an important case giving valuable insights into future challenges. It gives important insights into workload, different ways of working and also preprints that became an issue when media picked up unverified data and never followed up on the studies once peer reviewed. It would be interesting to have your results compared to other countries. Do you know if any similar research have been done that your case can be compared to? I have only a few comments in the attachment.

Authors’ response:

Thank you for your seeing the value of our paper. We are not aware of similar studies in other countries, to date, but hope there are in future.

The attachment noted above was not provided with the original decision email. On the advice of Dr Quinn Grundy, (email dated 9 August 2021, reference: PLOS ONE: PONE-D-21-12411 - [EMID:fd725523776fcf73]), we have revised and submit this manuscript without responses to this feedback.

---

## [Editor Report · Decision Letter 1]

18 Aug 2021

Impacts of COVID-19 on clinical research in the UK: a multi-method qualitative case study

PONE-D-21-12411R1

Dear Dr. Wyatt,

We’re pleased to inform you that your manuscript has been judged scientifically suitable for publication and will be formally accepted for publication once it meets all outstanding technical requirements.

Kind regards,

Quinn Grundy, PhD, RN

Academic Editor

PLOS ONE
---

## [Editor Report · Acceptance letter]

23 Aug 2021

PONE-D-21-12411R1 

Impacts of COVID-19 on clinical research in the UK: a multi-method qualitative case study 

Dear Dr. Wyatt:

I'm pleased to inform you that your manuscript has been deemed suitable for publication in PLOS ONE. Congratulations! Your manuscript is now with our production department. 

Kind regards, 

on behalf of

Dr. Quinn Grundy 

Academic Editor

PLOS ONE